# The Improvement of Sensory and Bioactive Properties of Yogurt with the Introduction of Tartary Buckwheat

**DOI:** 10.3390/foods11121774

**Published:** 2022-06-16

**Authors:** Yuanyuan Ye, Pei Li, Jiaojiao Zhou, Jiangling He, Jie Cai

**Affiliations:** 1National R&D Center for Se-Rich Agricultural Products Processing, Hubei Engineering Research Center for Deep Processing of Green Se-Rich Agricultural Products, School of Modern Industry for Selenium Science and Engineering, Wuhan Polytechnic University, Wuhan 430023, China; yyyuan0127@163.com (Y.Y.); jiaojiaozhou@whpu.edu.cn (J.Z.); 2Key Laboratory for Deep Processing of Major Grain and Oil, Ministry of Education, Hubei Key Laboratory for Processing and Transformation of Agricultural Products, Wuhan Polytechnic University, Wuhan 430023, China; m13247172326@163.com

**Keywords:** yogurt, Tartary buckwheat, response surface methodology, sensory property, volatile compound

## Abstract

The incorporation of cereals in yogurt has recently gained increasing consumer approval, for its high nutritional value and health benefits, all over the world. Following this emerging trend, Tartary buckwheat (TB) was supplemented into yogurt as a natural functional ingredient in order to develop a yogurt with enhanced product characteristics and consumer acceptability. The impact of TB addition on physicochemical properties (pH, acidity, apparent viscosity, etc.) and the viability of lactic acid bacteria in yogurt was investigated. It is found that the TB introduction can reduce the pH, increase the acidity and apparent viscosity, and also greatly boost the bioactivities of yogurt. Response surface analysis demonstrated that yogurt with 8 g of TB, 10 g of sugar, and a fermentation duration of 5 h had the highest overall acceptability, and these cultural conditions were chosen as the best. Furthermore, the TB-added yogurt had not only a better sensory and aroma profile, but also good prospective health advantages when compared to regular yogurt. Our research shows that adding TB to yogurt has a significant positive impact on both overall quality and sensory characteristics, making a compelling case for using TB yogurt and developing new fermented dairy products.

## 1. Introduction

Yogurt, a nutrient-rich functional food manufactured from the lactic acid fermentation of milk at a specific fermentation temperature and time, is currently one of the most popular dairy products in the world [1]. Yogurt not only retains the excellent nutritional quality of raw milk, rich in carbohydrates, fats, proteins (casein), minerals (calcium and phosphorus), and vitamins (B vitamins), but also endows yogurt with a large number of new biological metabolites (enzymes, polypeptides, etc.) and volatile components due to the metabolism of fermenting microorganisms, and exhibits good digestibility [2,3]. Furthermore, a growing body of evidence shows that eating yogurt has a variety of health benefits, including improving lactose digestion, stimulating the immune system, controlling body weight, relieving constipation, and lowering the risk of intestinal diseases such as inflammatory bowel disease and colon cancer [4]. Rising consumer awareness and the pursuit of healthy lifestyles have resulted in a remarkable increase in demand for innovative milk-based fermented products containing bioactive molecules and promising intrinsic health benefits in recent years, which has been accelerated by lifestyle changes and growing economic development [5,6]. Many studies have shown that adding different ingredients to yogurt (e.g., rice bran, hazelnut slurry, organic green banana flour, passion fruit peel powder, and fruit fibers) can boost lactic culture viability and improve physicochemical and organoleptic features [2,7,8,9,10]. The introduction of natural foods into yogurt to boost its nutritional and health benefits has been an emerging trend.

Tartary buckwheat (*Fagopyrum tataricum* (L.) Gaertn., TB), a member of the *Polygonaceae* family, is a widely consumed functional food due to its superior economic, nutritional, and medicinal benefits [11]. TB is rich in amino acids, dietary fiber, vitamins and minerals, and is particularly known for its abundant flavonoids (rutin, quercetin, anthocyanin, etc.). These superiorities can boost immunity, prevent the cardiovascular diseases, improve anti-oxidation and anti-aging effects, and have hypoglycemic and hypo-lipidemic properties [12,13]. Yogurt contains almost no fiber or flavonoid, whereas the addition of TB to yogurt may increase bioactive substances and improve health functions [14]. The presence of biologically active compounds can have an effect on the viability of probiotics in yogurt [15,16]. Thus, the use of TB as a possible addition to yogurt before the fermentation process is highly valuable.

In this work, we determined the optimal fermentation parameters by evaluating the effects of the TB additive amount on the physicochemical properties and acidogenic bacteria activity in yogurt. Response surface analysis was used to assess the relationship between sensory score and culture conditions. In addition, the sensory properties and volatile profile of yogurt were evaluated through gas chromatography-mass spectrometry experiments and by electronic tongue assays in order to assess the potential effects of adding appropriate TB to yogurt. Moreover, a cell assay was carried out to investigate the cytotoxicity of TB yogurt in H_2_O_2_-induced IPEC-J2 cells. The aim of this research was to optimize the cultural conditions of yogurt added with TB in order to create a dairy product with desirable sensory and bioactive properties. This could have instructive significance in terms of the development and commercialization of functional yogurt.

## 2. Materials and Methods

### 2.1. Materials and Reagents

The whole milk powder was purchased from Yili Industrial Group Co., Ltd. (Hohhot, China). TB was purchased from Chengdu Jintiankang Food Factory (Chengdu, China) and ground into flour (200 mesh) by a mill (Bear Electric Appliance Co., Ltd., Foshan, China). Sodium hydroxide (analytical reagent, AR), sodium chloride (AR), phenolphthalein (AR), and hydrogen peroxide (H_2_O_2_) were purchased from Sinopharm Chemical Reagent Co., Ltd. (Beijing, China). De Man-Rogosa-Sharpe (MRS) and modified Chalmers (MC) medium were purchased from Qingdao Rishui Bio-technologies Co., Ltd. (Qingdao, China). IPEC-J2 cells were donated from the laboratory of food science and engineering in Wuhan Polytechnic University. Dulbecco’s modified eagle medium (DMEM), fetal bovine serum (FBS), and penicillin/streptomycin were bought from Gibco Company (Grand Island, NY, USA). Beyotime Company (Haimen, China) provided the 3-(4,5-dimehthylthiazol-2-yl)-2,5-diphenyltetrazolium bromide (MTT) cell proliferation kit and cytotoxicity assay kit.

### 2.2. Yogurt Preparation

At 65 °C, the whole milk powder (21.1% fat, 19.5% protein, 12 g) and sugar were dispersed in 100 mL of water. The milk was obtained via pasteurization at 90–95 °C for 20 min before being cooled to 42 °C. The fresh samples were then supplemented with the TB (0 g, 4 g, 6 g, 8 g, 10 g, or 12 g), followed by inoculation with 0.05% (*w*/*v*, g/mL) of strains (a mixture of *Lactobacillus bulgaricus* and *Streptococcus thermophilus*). The fermentation process was finished in an incubator at 42 °C. The as-prepared yogurt containing TB was stored in a refrigerator at 4 °C for further use. All the tests were concluded within 2 days of storage.

### 2.3. Physicochemical Determinations

A Starter-3100 laboratory pH meter (Ohaus Instruments, Shanghai, China) was used to measure the pH of yogurt samples during fermentation (0, 1, 2, 3, 4, and 5 h). The titratable acidity (TA, °T) was determined according to the Chinese National Standard GB 5413.34-2010. Phenolphthalein was used as an indicator, the yogurt sample (10 g) was mixed with deionized water (20 mL) and titrated with a 0.1 mol/L sodium hydroxide (NaOH) solution to an endpoint pH of 8.3. At a stirring speed of 750 rpm/min, the apparent viscosity of the yogurt containing TB was measured using an NDJ-7 Type Rotary Viscometer (Shanghai Precision and Scientific Instrument Co., Ltd., Shanghai, China). All the samples were assayed in triplicate.

### 2.4. Sensory Evaluation

Fresh yogurt (1st day of storage at 4 °C) was assessed for basic sensory attributes, including color, shine, texture, flavor, and taste. Sensory evaluation was conducted with twenty trained volunteers (10 male persons and 10 female persons aged 20–40 years old) familiar with sensory analysis of yogurt products by combining the guidelines from the industry norm (industry standard for China’s dairy industry RHB 104-2020). The samples (35 g of yogurt) presented in 100 mL of plastic cups were coded with random three-digit numbers and presented in random order. The panelists were asked to rinse their mouths with clean water and wait at least 1 min between different samples. To eliminate interference, the sensory assessment was conducted in separate booths with no communication allowed between participants. The total score was 100, and the average value was used to evaluate the sensory acceptance of the yogurt after the highest and the lowest scores were removed.

### 2.5. Microbiological Determination

*Lactobacillus bulgaricus* and *Streptococcus thermophilus* measurements were performed in accordance with the national food safety standard microbiological examination: Lactic acid bacteria test (GB 4789.35-2016). The yogurt samples were diluted to the required concentration in sterilized physiological salin. *Lactobacillus bulgaricus* strains were detached via using selective de Man-Rogosa-Sharpe (MRS) agar and cultured at 37 °C in an anaerobic atmosphere for 72 h. *Streptococcus thermophilus* strains were picked out under the aerobic culture environment on modified Chalmers (MC) medium at 37 °C for 72 h. Plates containing 30–300 colonies were selected to count the colonies (CFU/g).

### 2.6. Experimental Design

Response surface methodology (RSM) is a systematic experimental mathematical model for predicting sorption behavior and evaluating the relative importance and interaction of each parameter (TB concentration, sugar concentration, and fermentation time). To obtain a high-quality yogurt, RSM based on a three-factor and three-level Box-Behnken design (BBD) was employed to study the optimum fermentation conditions for TB-containing yogurt. The value of each factor was obtained based on the results from the previous single-factor experiment, and the score for sensory evaluation of the yogurt samples was the response value. This experimental design consisted of 17 runs in total, including 12 factor points and 5 replications at the center point to estimate the experimental error. The experimental design matrix and its results are shown in Table 1. The response factors were monitored with a second-order polynomial equation (Equation (1)) to figure out the predicted response (Y).
Y = β_0_ + β_1_A + β_2_Β + β_3_C + β_12_AΒ + β_13_AC + β_23_ΒC + β_11_A^2^ + β_22_Β^2^ + β_33_C^2^(1)
where Y is the response variable (sensory score), β_0_ is a constant, β_1_ and β_2_ are the linear coefficients, β_11_, β_22_, and β_33_ are the quadratic coefficients, and β_12_, β_13_, and β_23_ are the interaction coefficients. A, B, and C represent TB concentration, sugar concentration, and fermentation time, respectively. The BBD output also contains the 3D surface response plot, which indicates the interaction between the responses and independent variables.

### 2.7. Electronic Tongue Analysis

The sensory characteristics of yogurt samples were measured using the TS-5000Z E-tongue instrument (Insent Inc., Atsugi-shi, Japan). Eight basic tastes, including sourness, saltiness, richness, umami, aftertaste-A, aftertaste-B, astringency, and bitterness, were determined by diluting the yogurt with an equal volume of deionized water, and 40 mL of supernatant was added into the cup after centrifugation for 10 min at 10,000 rpm/min. Each assay was repeated three times.

### 2.8. Gas Chromatography-Mass Spectrometry (GC-MS) Analysis

The extraction of volatile compounds was performed by solid phase microextraction (SPME) using a DVB/CAR/PDMS (divinyl-benzene/carboxen/polydimethylsiloxane) fiber (50/30 μm thickness; Supelco, Bellefonte, PA, USA). A 15-g sample of yogurt was weighed into a 20 mL vial and sealed with a cap containing a polytetrafluoroethylene (PTFE)/silicone septum. The volatile compounds were extracted using a SPME fiber in the headspace at 55 °C for 40 min after pre-equilibration at 55 °C for 20 min [17]. The GC-MS analyses for the extracted volatile components were performed on the Agilent Technologies 7890GC/5975C instrument (Agilent Technologies, Santa Clara, CA, USA) coupled with the ionized energy of the mass selective detector (MSD) mode at 70 eV with a scanning mass range of 30–500 *m*/*z*. Volatile compounds absorbed into the SPME fiber were passed through a HP-FFAP capillary column (30 m length × 0.25 mm i.d., 0.25 μm film thickness; Agilent Technologies, USA). The heating program process is as follows: The temperature was first set at 40 °C and maintained for 2.5 min, then raised to 140 °C with a heating speed of 5 °C /min and held for 5 min, and finally heated to 230 °C at a rate of 6 °C /min and kept for 5 min. The temperatures for the ion source, injector, and transmission line were 230 °C, 250 °C, and 230 °C, respectively. Helium at a constant flow rate of 2.0 mL/min was used as a carrier gas through the column.

### 2.9. Bioactive Assay

To compare the biological activity of common yogurt and TB yogurt, we selected intestinal porcine epithelial cells (IPEC-J2 cells) isolated from unfed newborn piglet for our investigation, based on the importance of gut activities in the digestion and absorption of food [18]. The IPEC-J2 cells were cultured in DMEM medium containing 10% FBS and 1% antibiotics (penicillin/streptomycin) at 37 °C in a humidified incubator under 5% CO_2_. For the optimization of H_2_O_2_ concentration, 5 × 10^3^/mL cell suspension was seeded into 96-well plates with 200 μL per well, 6 wells in each group, which was allowed to adhere for 24 h before treatment, and then given final concentrations of 3.4, 3.6, 4, 4.2, 4.4, 4.6, 4.8, and 5 mmol/L H_2_O_2_ to mimic the oxidative damage process. The medium was then removed and a solution of 3-(4,5-dimehthylthiazol-2-yl)-2,5-diphenyltetrazolium bromide (MTT) (0.5 mg/mL in medium) was added. The incubation was kept at 37 °C for 4 h, and finally the culture medium was removed. Cell viability was performed by the MTT colorimetric assay as follows: the culture solution in each well was removed, and cells were washed twice with PBS buffer, then 200 μL of serum-free medium containing MTT solution (0.5 mg/mL) was added to each well for the incubation at 37 °C for 4 h. Then, 100 μL of dimethyl sulfoxide (DMSO) was introduced to dissolve the precipitate. All experiments were conducted in triplicate with appropriate blanks. The absorbance was recorded at 490 nm by an EnSpire Multimode Reader (Perkin Elmer, Waltham, MA, USA).

### 2.10. Statistical Analysis

GraphPad Prism 9 (GraphPad Software Inc., San Diego, CA, USA) and Origin 2019 (OriginLab, Northampton, MA, USA) were used for statistical analyses. RSM design and statistical analyses were conducted using Design-Expert Version 12.0 (Stat-Ease Inc., Minneapolis, MN, USA). All the data shown in this work were collected by at least three independent experiments.

## 3. Results

### 3.1. Effects of TB on pH, Acidity, and Apparent Viscosity of Yogurt

The appearance of common yogurt and TB yogurt is shown in Figure 1. It is clear that common yogurt appeared a milky white color, while TB yogurt exhibited a light yellow color and TB was well-dispersed in it.

The pH of yogurt is an important indicator of its quality. The effect of TB content (g) on pH is shown in Figure 1A. The pH values of all yogurt groups exhibited a clear decreasing trend over fermentation time, because lactic acid bacteria such as *Lactobacillus bulgaricus* and *Streptococcus thermophilus* can exhibit rapid reproduction during fermentation and ferment lactose to lactic acid [4]. The pH of yogurt decreased rapidly within 2 h and showed a positive correlation upon the introduction of TB. A slow decline was observed at the fermentation duration of 2–5 h. The results show that the pH was higher in the plain yogurt than that in the TB-treated yogurt. Furthermore, after 5 h of fermentation, the pH value of all treatments were in the range of 4.38–4.57, and yogurt (100 mL) with 10 g of TB showed the lowest pH value (4.38).

The acidity showed an overall increase in all groups over the fermentation time (Figure 1B), due to both sugar consumption and lactate accumulation acidifying the milk. The initial acidity difference among the adjacent groups was weak when the amount of TB was less than 10 g, and it significantly enlarged upon the addition of TB beyond 10 g. There was a rather slow increase in the titration acidity in the first 2 h of fermentation, followed by a fast-rising trend in the first 2 h and a gradually slowing down trend in the last 1 h. It was obvious that the acidity increased in response to the increasing TB concentration, and the yogurt finally displayed remarkable acidity differences with the introduction of TB.

The apparent viscosity is an extremely important factor in evaluating the yogurt, as it represents the consistency and stiffness of yogurt, and also reflects the taste, tissue shape, and stability [19]. The apparent viscosities of all yogurt samples increased over the fermentation time throughout the duration of 5 h (Figure 1C), which might be attributed to the structural rearrangement of yogurt protein gel and the water-retaining action of hydrophobic protein. The apparent viscosity of the yogurt samples barely changed among groups within the first 2 h. The most dramatic rise occurred after incubation for 2 h, and the apparent viscosity peaked at the end. The sample containing 8 g of TB had the highest apparent viscosity performance (0.68 Pa·s) after fermentation for 5 h. The apparent viscosity enhancement might be associated with the presence of fibers in the TB-added yogurt, which resulted in the improvement and rearrangement of the casein gel network structures as well as the suppression of shrinkage and whey expulsion [16]. Furthermore, the apparent viscosity of yogurt is associated to the production of acid. When the acidity rises, the protein in the milk forms a firmer gel, causing the higher apparent viscosity in yogurt [20]. It is also worth noting that the increasing trend of apparent viscosity gradually slowed down, most likely due to increasing TB addition.

These findings are in good agreement with previous studies on apples, wheat fibers and rice bran [21]. Previous research has found that probiotic yogurt samples incorporating fiber had greater viscosity and acidity, and lower pH and syneresis when compared with plain yogurts [7]. The considerable decrease in pH and the increase in acidity and apparent viscosity should be owing to the increased metabolic activities of fermentative bacteria, which initiate the accumulation of metabolites such as phenolics, flavonoids, and organic acids [22]. As a conclusion, these results suggest that the addition of TB to yogurt could improve the quality of the yogurt.

### 3.2. The Viability of Acidogenic Bacteria

The most important qualitative parameter in yogurt is the viability of acidogenic bacteria [23]. The number of fermentative bacteria, including *Lactobacillus bulgaricus* and *Streptococcus thermophilus*, which are known for fermenting sugar in milk into lactic and other organic acids, was measured in this assay. As shown in Figure 1D, the addition of TB contributed to improving the total viability of *Lactobacillus bulgaricus* and *Streptococcus thermophilus* in yogurt when compared to the control. The results showed that TB supplementation had an evident boosting impact on *Lactobacillus bulgaricus* colony counts, with the highest *Lactobacillus bulgaricus* count (18 × 10^5^ CFU/g) obtained after 5 h of fermentation in the sample containing 10 g of TB. Moreover, there was a dramatic reduction in the viable counts of *Lactobacillus bulgaricus* in yogurt with 12 g of TB addition compared with yogurt with 10 g of TB addition, which was likely to be due to an increase in the production of metabolites as a result of increased metabolic activity. The accumulation of metabolites has a specific inhibitory effect on cell growth, which can lead to a reduction in bacterial viability [24]. Simultaneously, the quantity of *Streptococcus thermophilus* declined gradually with the increase of TB addition, and yogurt with 4 g of TB addition (9 × 10^5^ CFU/g) had more than fourfold greater than plain yogurt (2 × 10^5^ CFU/g) and had the maximum amount of *Streptococcus thermophilus* among all samples.

In numerous circumstances, the health benefits of yogurt can be ascribed to the starter cultures (*Lactobacillus*
*bulgaricus* and *Streptococcus thermophilus*) [25]. Previous work demonstrated that a number of factors influence the survival of fermentative bacteria in yogurt, including formulation factors (pH and total titratable acidity, hydrogen peroxide, molecular oxygen, strains of probiotic bacteria and microbial interactions, ripening factors, salt, growth promoters, and food additives), process factors (types of inoculation, heat treatment, incubation temperature, and storage temperature), and packaging [26]. TB is a nutrient-dense food that is abundant in flavonoids, dietary fibers, proteins, vitamins, and minerals. This indicates that TB could be used as a rich growth medium for yogurt bacteria to promote its growth and metabolic processes. This is a significant finding because increasing the amount of viable bacteria in yogurt can improve the rate of protein digestion and nutritional value [27].

### 3.3. Effect on Yogurt Sensory Score

Sensory analysis is one of the most extensively utilized methodologies for quality characterization, and it is crucial in determining consumer approval of yogurt products. The bar plot depicted the effect of different formulations (TB and sugar) additions and fermentation duration on the sensory score of yogurt samples (Figure 2), and this was presented in the form of average scores from yogurt samples that were documented by a trained panel.

The sensory quality of yogurt was greatly affected by the incorporation of TB. It was discovered in this study that when the amount of added TB increased, the sensory score went up at first, then reduced. The yogurt that had 8 g of TB added to it obtained the highest score of 92, which means that there was a considerable overall improvement in the yogurt samples’ acceptability.

As the amount of sugar added to yogurt increased, the sensory score improved significantly. The highest sensory score was achieved by adding 10 g of sugar to yogurt. However, there was a trend toward a dramatic drop in the sensory quality as the amount of sugar added went up. This is because too much sugar can negatively affect the overall taste [28].

The sensory quality of dairy products can be strongly influenced by duration of fermentation. The sensory score of the yogurt rose substantially when the fermentation time was between 4–5 h, but decreased when the time was extended to 6 h. On the one hand, choosing the appropriate fermentation time can help to boost flavor. Acidogenic bacteria, on the other hand, may keep making acid and causing an imbalance in the sugar/acid ratio if the fermentation process goes on for a long time.

Therefore, the optimum TB and sugar additions and fermentation time ranges for the succeeding experiments were validated as 6–10 g, 8–12 g, and 4.5–5.5 h, respectively. With these restrictions in place, the treated yogurt had a much more acceptable texture, taste, and overall preference.

### 3.4. Fitness Verification of RSM Models and Optimization

BBD matrix and response values are listed in Table 1, and the results from the analysis of variance (ANOVA) are provided in Table 2. A multinomial regression analysis was used to assess the relationship between sensory score and observed variables, and the second-order polynomial model was mathematically expressed in the following Equation (2):Y = −1044.1 + 68.0A + 16.2Β + 327.3C + 0.4AΒ + 0.5AC + (1.7 ×10^−14^) ΒC − 4.8A^2^ − 1.0Β^2^ − 34.0C^2^(2)

The regression results indicated that the model was statistically significant (*p* < 0.01) and had an F value of 13.47. The predicted value and actual value were found to be in good agreement (R^2^ = 0.95). The adjusted-R^2^ was 0.87, demonstrating that only 12.48% of the variance could not be predicted by this model. The nonsignificant lack-of-fit further suggests that this model showed a range of satisfaction up to good accuracy when it was used to anticipate the relevant responses. The adequate precision value was 10.27, revealing a significantly adequate signal-to-ratio relationship. The coefficient of variation (C.V.) of yogurt sensory score was 2.83% (<5.00%), indicating high reproducibility. According to the *p* value, the linear effect of C, as well as the square effects of A (A^2^) and C (C^2^), were extremely significant, showing that the level of 1%, and B^2^ was significant at the 5% level, and indicating that the amount of fermentation time had a significant correlation with the sensory score.

In order to facilitate the interpretation of the key interactions and gain a better understanding of the complex linkages among the model variables, three response surface plots were figured as shown in Figure 3. The final results showed that the optimal levels of addition formulation and fermentation duration were 8 g of TB, 10 g of sugar, and 5 h of fermentation time, respectively.

### 3.5. Effect of TB Addition on Taste Attributes of Yogurt

The E-tongue converts an electrical signal into a taste signal to distinguish the taste of yogurt, and it has a low threshold of sensation and avoids sensory evaluation subjectivity [29]. Figure 4 shows the response values for eight different taste attributes (sourness, saltiness, richness, umami, aftertaste-A, aftertaste-B, astringency, and bitterness). The signal values of saltiness, richness, umami, and aftertaste-B of the yogurt increased considerably after TB was added to the milk before the fermentation process, but the signals of bitterness and aftertaste-A only increased slightly. Compared with the common yogurt, the response value of astringency in optimal yogurt with TB addition (TB yogurt) was slightly lower. There was also a marked decrease in the level of sourness in TB yogurt. Thus, TB can greatly improve the taste profile of yogurt, which could be due to the abundance of flavonoids, organic acids, phenolic acids, and sugars in TB.

### 3.6. Effect of TB on Volatile Flavor Profile of Yogurt

Aroma is one of the most fundamental and important characteristics of yogurt, which can affect the preference and acceptability of consumers. The production of lactic acid and a complex mixture of volatile compounds promote the formation of a distinctive flavor in yogurt [30]. Table 3 shows the relative contents of volatile components in yogurt. The addition of TB altered the volatile compound profile of the products compared to ordinary yogurt, as there was a considerable variation in the amount of most volatile compounds after the TB treatment. In this assay, a total of 42 unique volatile compounds, including aldehydes, hydrocarbons, alcohols, esters, and others, were effectively identified in yogurt samples. A stacked column histogram (Figure 5) is pr3eenteded based on the percentage contribution of each kind of volatile compound among the total peak areas, and provided an efficient way to compare the volatile flavor profile of yogurt with different production processes. Aldehydes were the most abundant flavor substances in both common yogurt and TB yogurt, followed by hydrocarbons and alcohols. Additionally, it was easy to find that the corporation of TB declined the relative percentage of esters and enhanced the relative ratio of aldehydes, acids, alkenes, and ketones, with considerable consequences for the aroma of yogurt.

Aldehydes, which are mostly generated by the oxidation of unsaturated fatty acids, are the essential components that have an impact on the aroma of yogurt [17]. It is apparent that hexanal, heptanal, octanal, nonanal, decanal, undecanal, dodecanal, and 2-dodecenal are important contributors to yogurt aroma formation. Jane et al. reported that hexanal, nonanal, decanal, and (*E*,*E*)-2,4-decadienal are the main characteristic aroma compounds of Tartary buckwheat [31]. During the fermentation process, hexanal can yield an unpleasant green and grassy odor and have a negative impact on sensory quality. There was a decrease in the hexanal content of yogurt after the addition of TB. Nonanal is one of the primary components that contribute to yogurt’s overall flavor, and the level of nonanal in yogurt with the introduction of TB or not appeared to be equal (17.07% and 17.08%, respectively). Decanal has previously been reported to be associated with the fatty aroma of fermented dairy products [32]. Benzaldehyde, an important aromatic compound in yogurts, can emit a bitter almond odor at a low concentration and maraschino cherry notes at a high concentration [33]. 2-Dodecenal has a strong fat-like smell and taste, while it imparts a citrus aroma after the dilution treatment. Though conventional yogurt had 6.45% of 2-dodecenal, it was not detected in TB yogurt. Notably, heptanal is a main contributor to the green and sweet aroma in TB yogurt [34], accounting for the highest volatile share (18.06%), which was more than three times that detected in regular yogurt (4.87%).

In regards to the hydrocarbon group, nine different hydrocarbons were identified from the yogurt’s volatile compounds. The addition of TB increased the relative percentages of undecane, dodecane, tridecane, tetradecane, pentadecane, 1,2-epoxy octadecane, and heptadecane in yogurt. In addition, traditional yogurt contained more cyclododecane than TB yogurt. Hydrocarbons are usually precursors for other volatile compounds and have no direct effect on the formation of the characteristic aroma in the products [32].

Three types of alcohols, including *trans*-2-methylcyclopentanol, 2-cyclohexene-1-ol, and nonadecanol, were identified during the process of the oxidation of lipids, and the reduction of various aldehydes and ketones, where they are usually associated with floral, fruity and alcoholic odors in dairy products [35]. The TB yogurt showed a lower concentration of *trans*-2-methylcyclopentanol, a higher concentration of nonadecanol, and an equal concentration of 2-cyclohexene-1-ol compared to the control. Despite the fact that most alcohols are used as intermediates in the synthesis of esters, they can lessen the sour taste that acid makes [36].

In the volatile fraction of TB yogurt, four ester compounds were detected, including methyl palmitate, methyl linoleate, methyl oleate, and methyl stearate, with varying degrees of concentration reduction in comparison with plain yogurt.

Acids are key volatile compounds in yogurt and serve as precursors for the synthesis of alcohols and esters [37]. 5-Hexenoic acid was detected exclusively in ordinary yogurt, but pentadecanoic acid was not present. The palmitic acid content (1.86%) in TB yogurt was greater than that in the control (1.21%).

Ketones, which contribute to the characteristic aroma of many foods, are mostly from oxidation, thermal degradation, and amino acid degradation products of unsaturated fatty acids [38]. The relative percentage of 2-nonanone in TB yogurt (1.27%) was higher than that in normal yogurt (0.44%). According to Liu et al. [36], the presence of 2-nonanone was also capable of weakening pungent flavor.

As for heterocycles, 2-pentylfuran (caramel notes and nut odor) was identified in both yogurt groups, and its concentration rose slightly when TB was added. Yogurts with TB addition or not had the same relative proportion of amyl-benzene in this experiment.

This suggests that adding TB to yogurt had a direct effect on the production of volatile aromatic compounds. Such an effect was most likely to be linked to a significantly higher level of total acidogenic bacteria counts in TB yogurt compared to common yogurt, due to bacteria’s critical roles in altering the composition and concentration of volatile components in the milk base [39].

### 3.7. IPEC-J2 Cell Viability Affected by TB Yogurt

IPEC-J2 cells were exposed for 24 h to varying concentrations of H_2_O_2_ (3.6–5.0 mmol/L). The results showed that H_2_O_2_ caused significant cytotoxicity and induced cell death in a dose-dependence manner (3.6 mmol/L and 5 mmol/L H_2_O_2_ induced oxidative damage of 32.76% and 90.90% of the cells, respectively, Figure 6A). A concentration of 3.6–4.2 mmol/L was chosen as the optimal culture condition for the subsequent experiment. The IPEC-J2 cells were divided into two groups: the control group (marked as control) and the intervention group (cultured with TB yogurt supernatant, marked as TB yogurt). As compared to the H_2_O_2_ group, the survival rate of cells went up dramatically when they were pre-treated with the supernatant of TB yogurt before H_2_O_2_, and reached a low point with the treatment of 3.8 mg/L of H_2_O_2_ (Figure 6B). The TB yogurt pretreatment significantly attenuated H_2_O_2_-induced cytotoxicity. As a result, H_2_O_2_ can cause potent cytotoxicity to IPEC-J2 cells, which was significantly reversed by TB yogurt.

## 4. Conclusions

The addition of Tartary buckwheat (TB) ranging from 4–12 g had a positive effect on the physicochemical and textural properties of yogurt, including a decrease in pH, a rise in titratable acidity, and an increase in apparent viscosity. It was also shown that adding TB to yogurt dramatically promoted the growth of *Lactobacillus Bulgaricus* and *Streptococcus Thermophilus*. The response surface methodology (RSM) was confirmed to be an effective and powerful approach to optimize fermentation conditions and evaluate independent factors (TB content, sugar content, and fermentation time) and their interaction effects on yogurt sensory score. There was excellent agreement between predictions and observations of sensory score in the present study. The yogurt (100 mL) with 8 g of TB, 10 g of sugar, and a 5 h of fermentation duration had the highest overall acceptability, and therefore were chosen as the optimal culture conditions. Additionally, TB yogurt prepared under optimal conditions exhibited not only better sensory characteristics but also excellent potential health benefits in comparison with common yogurt. Therefore, TB yogurt is a promising option to develop functional yogurt rich in volatile components and health-promoting compounds, and can offer multiple and commercial production uses in fermented dairy products. This study can further be extended to derive insights into the underlying mechanism of interactions among the chemical constituents of TB yogurt.

## Data Availability

The data presented in this study are available on request from the corresponding author. The data are not publicly available due to privacy.

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
