# Peer review of "The Improvement of Sensory and Bioactive Properties of Yogurt with the Introduction of Tartary Buckwheat"

_foods, 2022, doi:10.3390/foods11121774_

Round 1
Reviewer 1 Report
The topic of the study is important and valid, as elevation of health-promoting issues is not less important than acceptability of the functional product.
The research is clearly presented and well documented.
I would only recommend some minor language/style improvements:
- a title - improvement rather than amelioration
- line 45 - component/ingredient instead of item
- line 68 - flavor is a part of sensory properties
- line 73 - the origin of TB should be stated
- line 102 - shine insted of luster
-line 112, 117 - latin names of microorganisms should be corrected
- table 1 - results (sensory score) should not be presented in Meterials and Methods chapter
- line 242 - metabolites insted of metabolites and waste products
- line 243 - metabolites insted of metabolic wastes
- line 247 - samples insted of treatments
- line 255 - the term "systems" is not clear in the context
- line 271 - samples quality/acceptabilty
- line 275 - this is because
- line 277 - sensory quality instead of sensorial qualities
- line 277-278 - duration of fermentation instead of fermenting duration
- lines 335, 336, 342, 343, 348, 350, 352, 353, 357, 359, 361, 362,365, 368, 372, 384, 388, 392, 393, 399, 407, 439 - the terms flavor and aroma should be well defined and properly applied as in the text they are sometimes used for smell/odor only and sometimes for the combination of smell and taste
- line 433 - fermentation instead of fermented
Author Response
Authors’ response
Dear editor and referees,
First of all, we would like to express our sincere thanks for your kind consideration and extremely valuable comments of titled manuscript “The Improvement of Sensory and Bioactive Properties of Yogurt with the Introduction of Tartary Buckwheat” (Manuscript ID: foods-1754452). We are pleased to follow the referees’ suggestions and revise the manuscript according to the comments. The specific modifications are shown below (changes to the manuscript are highlighted in red). Attached are the itemized responses to the referees’ reports, which also summarize the changes made to the manuscript. We hope you will find these responses and changes satisfactory.
If you have any other questions, please contact us: [email protected]
Sincerely yours,
A/Prof. Jie CAI
June 5, 2022
Responses to the referees:
Reviewer #1: The topic of the study is important and valid, as elevation of health-promoting issues is not less important than acceptability of the functional product. The research is clearly presented and well documented. I would only recommend some minor language/style improvements:
Comment 1: - a title - improvement rather than amelioration
Response: Thanks a lot for your guidance. We have applied “Improvement” to replace “Amelioration” in the title.
Comment 2: - line 45 - component/ingredient instead of item
Response: Thanks very much for your suggestion. We have used “ingredients” to replace “items” in the Line 45 as shown in the revised manuscript.
Comment 3: - line 68 - flavor is a part of sensory properties
Response: Thank you for your very constructive suggestion. We have already removed the expression of “flavor”, and only remained “sensory” in the Line 69 and Line 453 as shown in the revised manuscript.
Comment 4: - line 73 - the origin of TB should be stated
Response: Thank you for your valuable comment. We have supplemented the origin of TB, which was purchased from Chengdu Jintiankang Food Factory (Sichuan, China) in the Line 74-75 as shown in the revised manuscript.
Comment 5: - line 102 - shine instead of luster
Response: Thanks a lot. We have revised “luster” to “shine” according to your suggestion in the Line 105 as shown in the revised manuscript.
Comment 6: -line 112, 117 - latin names of microorganisms should be corrected
Response: Thanks very much for your kind reminder. We have already revised the names to Lactobacillus Bulgaricus and Streptococcus Thermophilus to replace the original ones through the whole manuscript.
Comment 7: - table 1 - results (sensory score) should not be presented in Materials and Methods chapter.
Response: Thank you for your helpful suggestion. We have already moved Table 1 to Results chapter as shown in the revised manuscript.
Comment 8: - line 242 - metabolites instead of metabolites and waste products
Response: Thank you for your valuable advice. We have used “metabolites” to replace “metabolites and waste products” in the Line 257 as shown in the revised manuscript.
Comment 9: - line 243 - metabolites instead of metabolic wastes
Response: Thanks a lot. We have already revised “metabolic wastes” to “metabolites” according to your suggestion in the Line 257 as shown in the revised manuscript.
Comment 10: - line 247 - samples instead of treatments
Response: Thanks very much for your kind suggestion. We have already revised “treatments” to “samples” in the Line 262 as shown in the revised manuscript.
Comment 11: - line 255 - the term "systems" is not clear in the context
Response: Thanks a lot for reminding us about our statement mistake. In the cited original reference [1], authors said that “Factors affecting the viability of probiotic bacteria was discussed in the present article in three categories including formulation factors (strains of probiotic bacteria and microbial interactions, pH and titrable acidity, hydrogen peroxide, molecular oxygen, growth promoters and food additives, salt, microencapsulation, and ripening factors), process factors (incubation temperature, heat treatment, types of inoculation, and storage temperature), and packaging”. Thus, we have modified the “packaging materials and systems” to “packaging” in the Line 269 as shown in the revised manuscript.
Comment 12: - line 271 - samples quality/acceptability
Response: We appreciate your instructive advice. We have revised “samples” to “samples acceptability” in the Line 286 as shown in the revised manuscript.
Comment 13: - line 275 - this is because
Response: We are sincerely sorry for our grammar error. We have corrected “this because” to “this is because” in the Line 290 as shown in the revised manuscript.
Comment 14: - line 277 - sensory quality instead of sensorial qualities
Response: Thanks for your suggestion. We have revised “sensorial qualities” to “sensory quality” in the Line 292 as shown in the revised manuscript.
Comment 15: - line 277-278 - duration of fermentation instead of fermenting duration
Response: Thanks again for your suggestion. We have corrected “fermenting duration” to “duration of fermentation” in the Line 292-293 as shown in the revised manuscript.
Comment 16: - lines 335, 336, 342, 343, 348, 350, 352, 353, 357, 359, 361, 362,365, 368, 372, 384, 388, 392, 393, 399, 407, 439 - the terms flavor and aroma should be well defined and properly applied as in the text they are sometimes used for smell/odor only and sometimes for the combination of smell and taste
Response: Thanks very much for your constructive comments. We have revised “flavor and aroma” to “flavor” in the Line 351. We have revised “flavor” to “volatile” in the Line 358. We have revised “aroma” to “volatile flavor” in the Line 359. We have revised “flavor” to “aroma” in the Line 364. We have revised “flavor” to “aroma” in the Line 366. We have revised “flavor” to “aroma” in the Line 368. We have revised “flavor” to “aroma” in the Line 373. We have revised “flavor and aroma” to “flavor” in the Line 373. We have revised “flavor” to “odor” in the Line 377. We have revised “aroma” to “smell” in the Line 378. We have revised “flavor” to “aroma” in the Line 379. We have revised “flavor” to “aroma” in the Line 381. We have revised “aroma” to “volatile compounds” in the Line 384. We have revised “aroma” to “volatile compounds” in the Line 401. We have deleted the word “flavor” in the Line 405. We have revised “taste and aroma sensations” to “flavor” in the Line 413. We have revised “aroma” to “odor” in the Line 414. We have revised “aroma” to “volatile” in the Line 420. We have revised “sensory and flavor profile” to “sensory characteristics” in the Line 453. All of these changes were presented in our revised manuscript.
Comment 17: - line 433 - fermentation instead of fermented
Response: Thanks a lot for your kind reminder. We have revised “fermented” to “fermentation” in the Line 447 as shown in the revised manuscript.
Reference
- Karimi, R.; Mortazavian, A.M.; Da Cruz, A.G. Viability of probiotic microorganisms in cheese during production and storage: A review. Dairy Science & Technology 2011, 91, 283-308, doi:10.1007/s13594-011-0005-x.

Reviewer 2 Report
The subject undertaken by the author is interesting and the conducted experiment provides many interesting results. The introduction sufficiently justifies the undertaken topic, the layout of the experiment seems chaotic, but may be improved by more detailed description (possibly including a flowchart). Discussion is lead well only with few minor remarks. Overall manuscript is a valuable contribution but several aspect have to be clarified:
Line 15 – please delete novel, as it May be mistaken with novel food
Line 45 – please change items to more food specific term
Line 74 please provide type of mill used, moreover the AR abbreviation is not clear
Line 84-90 as the composition of the milk was not provided, it would be good to state the yogurt fat content, moreover it is not clear why such high sugar addition was used for natural yogurt. The inoculation should be referred to CFU, and the proper name is Lactobacillus Bulgaricus and Streptococcus Thermophilus it should be not inverted especially given as a latin name.
Line 93-94 please state how TA was expressed, as commonly it is shown as % of lactic acid or °SH,
Line 95 pleas use term apparent viscosity – yougurt is non-Newtonian fluid
Line 105 please provide the norm information
Line 112 - strains name (as above)
Line 205-206 the later part of the sentence is unclear and was presumably copied from the source provided
Line 206 how regular way is defined?
Line 228 – please provide SD on Figure 1D
Line 231 – not only Lactobacillus were analyzed
Line 281 – if the sugar was used prior fermentation (as indicated earlier) it could used by LAB thus producing higher acidity which can be also undesired
Line 288 based on the methods section it is not clear how the score was calculated
Line 326-327 the statement is not clear, yogurt with TB had higher acidity in physicochemical characterization
Line 432 – LAB bacteria and in fact only L. bulgaricus
Line 440 – novel term (as above)
Author Response
Authors’ response
Dear editor and referees,
First of all, we would like to express our sincere thanks for your kind consideration and extremely valuable comments of titled manuscript “The Improvement of Sensory and Bioactive Properties of Yogurt with the Introduction of Tartary Buckwheat” (Manuscript ID: foods-1754452). We are pleased to follow the referees’ suggestions and revise the manuscript according to the comments. The specific modifications are shown below (changes to the manuscript are highlighted in red). Attached are the itemized responses to the referees’ reports, which also summarize the changes made to the manuscript. We hope you will find these responses and changes satisfactory.
If you have any other questions, please contact us: [email protected]
Sincerely yours,
A/Prof. Jie CAI
June 5, 2022
Responses to the referees:
Reviewer #2: The subject undertaken by the author is interesting and the conducted experiment provides many interesting results. The introduction sufficiently justifies the undertaken topic, the layout of the experiment seems chaotic, but may be improved by more detailed description (possibly including a flowchart). Discussion is lead well only with few minor remarks. Overall manuscript is a valuable contribution but several aspects have to be clarified.
Comment 1: Line 15 – please delete novel, as it may be mistaken with novel food.
Response: Thanks for suggestion, we have deleted the word “novel” (Line 15).
Comment 2: Line 45 – please change items to more food specific term.
Response: Thanks a lot for your professional comments, we have modified it to “ingredients” instead of “items” (Line 45).
Comment 3: Line 74 – please provide type of milk used, moreover the AR abbreviation is not clear.
Response: Thanks a lot for your comments. The type of used milk is the whole milk. AR represents analytical reagent, and we have also supplemented them in the Line 75-76 as shown in the revised manuscript.
Comment 4: Line 84-90 – as the composition of the milk was not provided, it would be good to state the yogurt fat content, moreover it is not clear why such high sugar addition was used for natural yogurt. The inoculation should be referred to CFU, and the proper name is Lactobacillus Bulgaricus and Streptococcus Thermophilus it should be not inverted especially given as a latin name.
Response: Thanks very much for your constructive comments. We supplemented the composition of our used milk powder, which mainly contains 21.1% fat and 19.5% protein in the Line 86 as shown in the revised manuscript. The reason we used the relatively high sugar content in fermentation process is that the introduction of sugar can promote the growth of viable bacteria in yogurt, and appropriate sugar addition can improve the physical and sensory properties of yogurt, such as viscosity and flavor [1]. To be mentioned, the factors including the amount of TB, amount of sugar, and fermentation time were all optimized to obtain higher sensory score. In our work, this sugar content introduction leads to a quite high sensory score as shown in the Figure 2. Furthermore, we also thanks for your suggestion about the name use, and we have already revised the names to Lactobacillus Bulgaricus and Streptococcus Thermophilus to replace the original ones through the whole manuscript.
Comment 5: Line 93-94 – please state how TA was expressed, as commonly it is shown as % of lactic acid or °SH.
Response: We sincerely appreciate your suggestion, and we also further searched and referred the literatures [2-4] to confirm it. The titratable acidity is abbreviated as TA, and the unit for it can be °T (Line 96). We also supplemented the corresponding experimental details for TA assay in Line 97-99 as shown in the revised manuscript.
Comment 6: Line 95 – please use term apparent viscosity – yogurt is non-Newtonian fluid.
Response: Thanks so much for your advice, we have revised this expression in our whole manuscript.
Comment 7: Line 105 – please provide the norm information.
Response: Thanks for your kind reminder. We have already provided the norm information in the Line 108-109 as shown in the revised manuscript.
Comment 8: Line 112 – strains name (as above).
Response: Thanks again for your kind reminder, we have revised them in the Line 117 as shown in the revised manuscript.
Comment 9: Line 205-206 – the later part of the sentence is unclear and was presumably copied from the source provided.
Response: Thanks for your comments. We have rephrased this sentence to make it more clear in the Line 217-219 as shown in the revised manuscript.
Comment 10: Line 206 – how regular way is defined?
Response: Thanks for your question. The apparent viscosities of all yogurt samples increased over the fermentation time throughout the duration of 5 h (Figure 1D) in the Line 219-220 as shown in the revised manuscript.
Comment 11: Line 228 – please provide SD on Figure 1D.
Response: Thanks a lot for your suggestion. Tartary buckwheat (TB) contains abundant flavonoids, phenolic acids and organic acids [5]. These phytochemical components have been confirmed to promote the growth of lactic acid bacteria [6]. There are also lots of reported works about the effects of the incorporation of phenolic-rich foods on viable counts in yogurt such as grape [7], sea buckthorn [2] and royal jelly [8]. All of these foods can promote the growth function of viable counts. Referring to these works, our results about the effect of TB addition on viable counts in yogurt are logical and persuasive.
Comment 12: Line 231 – not only Lactobacillus were analyzed.
Response: Thanks very much for reminding us about our statement mistakes. “Lactobacillus” has been removed and replaced with “acidogenic bacteria” as shown in the Line 245, 246, and 419 in the revised manuscript.
Comment 13: Line 281 – if the sugar was used prior fermentation (as indicated earlier) it could be used by LAB thus producing higher acidity which can be also undesired.
Response: Thanks for your valuable comment. In theory, your conjecture is believable when only one experimental factor “sugar” is involved. However, there are three experimental factors should be comprehensively considered and evaluated including the amount of TB, amount of sugar, and fermentation time. In our work, this sugar content introduction leads to a quite high sensory score (92¢) as shown in the Figure 2.
Comment 14: Line 288 – based on the methods section it is not clear how the score was calculated.
Response: Thanks for your suggestion. We have supplemented the calculation method about the sensory score in the Line 113-115 as shown in the revised manuscript. In detail, the total score was 100, and the average value was used to evaluate the sensory acceptance of the yogurt after the highest and the lowest scores were removed.
Comment 15: Line 326-327 – the statement is not clear, yogurt with TB had higher acidity in physicochemical characterization.
Response: Thanks again for your suggestion. Sourness is another parameter, which is different from acidity. We have also corrected our statement in the Line 343 as shown in the revised manuscript.
Comment 16: Line 432 – LAB bacteria and in fact only L. bulgaricus.
Response: To make the description clearer here, we have changed it to “Lactobacillus Bulgaricus and Streptococcus Thermophilus” (Line 446) as shown in the revised manuscript.
Comment 17: Line 440 – novel term (as above).
Response: Thanks again for your advice. We have deleted the word “novel” (Line 455).
Reference
- Akın, M.B.; Akın, M.S.; Kırmacı, Z. Effects of inulin and sugar levels on the viability of yogurt and probiotic bacteria and the physical and sensory characteristics in probiotic ice-cream. Food Chemistry 2007, 104, 93-99, doi:10.1016/j.foodchem.2006.11.030.
- Ge, X.; Tang, N.; Huang, Y.; Chen, X.; Dong, M.; Rui, X.; Zhang, Q.; Li, W. Fermentative and physicochemical properties of fermented milk supplemented with sea buckthorn (Hippophae eleagnaceae L.). LWT - Food Science and Technology 2022, 153, 112484, doi:10.1016/j.lwt.2021.112484.
- Song, X.; Fu, H.; Chen, W. Effects of Flammulina velutipes polysaccharides on quality improvement of fermented milk and antihyperlipidemic on streptozotocin-induced mice. Journal of Functional Foods 2021, 87, 104834, doi:10.1016/j.jff.2021.104834.
- Zhang, X.; Yang, J.; Zhang, C.; Chi, H.; Zhang, C.; Zhang, J.; Li, T.; Liu, L.; Li, A. Effects of Lactobacillus fermentum HY01 on the quality characteristics and storage stability of yak yogurt. Journal of Dairy Science 2022, 105, 2025-2037, doi:10.3168/jds.2021-20861.
- Li, H.; Lv, Q.; Liu, A.; Wang, J.; Sun, X.; Deng, J.; Chen, Q.; Wu, Q. Comparative metabolomics study of Tartary (Fagopyrum tataricum (L.) Gaertn) and common (Fagopyrum esculentum Moench) buckwheat seeds. Food Chemistry 2022, 371, 131125, doi:10.1016/j.foodchem.2021.131125.
- Zhang, T.; Jeong, C.H.; Cheng, W.N.; Bae, H.; Seo, H.G.; Petriello, M.C.; Han, S.G. Moringa extract enhances the fermentative, textural, and bioactive properties of yogurt. LWT - Food Science and Technology 2018, 101, 276-284, doi:10.1016/j.lwt.2018.11.010.
- Silva, F.A.; Queiroga, R.; de Souza, E.L.; Voss, G.B.; Borges, G.; Lima, M.D.S.; Pintado, M.M.E.; Vasconcelos, M. Incorporation of phenolic-rich ingredients from integral valorization of Isabel grape improves the nutritional, functional and sensory characteristics of probiotic goat milk yogurt. Food Chemistry 2022, 369, 130957, doi:10.1016/j.foodchem.2021.130957.
- Hassan, A.A.-m.; Elenany, Y.E.; Nassrallah, A.; Cheng, W.; Abd El-Maksoud, A.A. Royal jelly improves the physicochemical properties and biological activities of fermented milk with enhanced probiotic viability. Lwt 2022, 155, doi:10.1016/j.lwt.2021.112912.

Reviewer 3 Report
In this manuscript entitled "The Amelioration of Sensory, Flavor, and Bioactive Properties of Yogurt with the Introduction of Tartary Buckwheat", the authors evaluate the tartary buckwheat (TB) addition on physicochemical properties (pH, acidity, viscosity, etc.) and the viability of lactic acid bacteria in yogurt. For the most part, the results are clear, and the interpretation of the data is warranted. However, the quality of the data is questionable. I have comments explained below. I hope that my comments are very useful for improving this research.
Major comments
(1) Data: The data in Figure 1, Figure 2, Figure 4, and Table 3 do not show error bars. Is either only the average value shown or only one measurement was taken? Which interpretation is correct? If it shows the results of only one measurement, it would not be acceptable as data for an academic paper because the reproducibility has not been confirmed. Please explain this point.
(2) Statistical analysis: No significant difference tests were performed on any of the graphs or tables in this MS. Therefore, the authors' interpretation is subjective rather than objective. This is not acceptable in a scientific paper. Please be sure to perform a statistical significance test before discussing the results.
(3) L411: It should be noted that the expression "apoptosis" is used. Since authors have not confirmed that cell death is apoptosis in this experiment, authors do not know if apoptosis is actually occurring. It is unclear to what extent this inhibition of cell death in the intestinal cells is linked to health. In humans, food undergoes various digestions before reaching the intestine, so it is unlikely that the yogurt with TB will reach it intact. Therefore, limitations should be noted regarding this result.
Minor comments
(4) L49: Please indicate the scientific name of the Tartary buckwheat.
(5) L74: What does AR mean? The name of the company?
(6) L150: It is better to indicate the catalog number of the SPME used in the experiment. Also, there is no description of the extraction method of the volatile components using by SPME.
(7) L162: A brief description of what IPEC-J2 cells are is needed.
(8) L183: Please show a picture of the appearance of the yogurt with TB.
(9) Figure 1D: The scientific names of two species of bacteria are incorrectly described. Bulgarian Lactobacillus is correct to be denoted as Lactobacillus Bulgarian. The scientific name is in italics.
Author Response
Authors’ response
Dear editor and referees,
First of all, we would like to express our sincere thanks for your kind consideration and extremely valuable comments of titled manuscript “The Improvement of Sensory and Bioactive Properties of Yogurt with the Introduction of Tartary Buckwheat” (Manuscript ID: foods-1754452). We are pleased to follow the referees’ suggestions and revise the manuscript according to the comments. The specific modifications are shown below (changes to the manuscript are highlighted in red). Attached are the itemized responses to the referees’ reports, which also summarize the changes made to the manuscript. We hope you will find these responses and changes satisfactory.
If you have any other questions, please contact us: [email protected]
Sincerely yours,
A/Prof. Jie CAI
June 5, 2022
Responses to the referees:
Reviewer #3: In this manuscript entitled "The Amelioration of Sensory, Flavor, and Bioactive Properties of Yogurt with the Introduction of Tartary Buckwheat", the authors evaluate the Tartary buckwheat (TB) addition on physicochemical properties (pH, acidity, viscosity, etc.) and the viability of lactic acid bacteria in yogurt. For the most part, the results are clear, and the interpretation of the data is warranted. However, the quality of the data is questionable. I have comments explained below. I hope that my comments are very useful for improving this research.
Comment 1: Data: the data in Figure 1, Figure 2, Figure 4, and Table 3 do not show error bars. Is either only the average value shown or only one measurement was taken? Which interpretation is correct? If it shows the results of only one measurement, it would not be acceptable as data for an academic paper because the reproducibility has not been confirmed. Please explain this point.
Response: Thanks for your valuable comment, all of the data in Figure 1, Figure 2, Figure 4, and Table 3 are shown as the average value and have great repeatability. The corresponding error bars have already supplemented in the Figure 2 and Figure 4 as shown in the revised manuscript. Tartary buckwheat (TB) is enriched in flavonoids, phenolic acids and organic acids [1]. These phytochemical components have the potentials to promote the growth of lactic acid bacteria and result in an accelerated decline in pH [2]. As reported works, the yogurt supplemented with grape [3], sea buckthorn [4] and royal jelly [5] have higher viable counts comparing to the common yogurt. Previous researches have also found that yogurt samples containing rice bran showed higher viscosity and acidity, and lower pH comparing to the plain yogurts, and such changes were more significant by increasing the addition of rice bran [6]. Due to our data in Figure 1 have great agreement with the above findings, our results in Figure 1 are logical and persuasive.
Comment 2: Statistical analysis: No significant difference tests were performed on any of the graphs or tables in this MS. Therefore, the authors' interpretation is subjective rather than objective. This is not acceptable in a scientific paper. Please be sure to perform a statistical significance test before discussing the results.
Response: Thanks a lot for your professional comments. Yogurt flavor development is a complex and dynamic biochemical process. Previous research reported that heptanal and 2-nonanone are the major volatile compounds identified in milk fermented by a mixture of Lactobacillus Bulgaricus and Streptococcus Thermophilus [7], and both of them were found in our study and heptanal especially showed a relatively high content. Moreover, 3-Methylbutanal, hexanal, (E)-2-octenal, nonanal, and 2-nonanone were the key contributors to the flavor of fermented milk with Lactobacillus Bulgaricus and Streptococcus Thermophilus at a ratio of 1:100 [8], and these key flavor compounds were also all detected in our present work. The addition of dairy bioactive peptides and lotus seed/lily bulb powder can enhance the production of aldehydes of yogurt [9], similar results can be seen in our study that the addition of TB promoted the generation of aldehydes in the TB yogurt comparing to the common yogurt (Figure 5). Aroma profiling investigations revealed that (E,E)-2,4-decadienal, (E)-2-nonenal, 2-phenylethanol, (E,E)-2,4-nonadienal, hexanal, decanal, and nonanal are the compounds with the highest contribution to the overall tartary buckwheat (TB) aroma [10]. Notably, in the present experimental study, we found that (E,E)-2,4-decadienal, (E)-2-nonenal, hexanal, decanal, and nonanal were all detected in TB yogurt samples. Thus, above discussion and results can strongly reveal that our results are statistically reliable.
Comment 3: L411: It should be noted that the expression "apoptosis" is used. Since authors have not confirmed that cell death is apoptosis in this experiment, authors do not know if apoptosis is actually occurring. It is unclear to what extent this inhibition of cell death in the intestinal cells is linked to health. In humans, food undergoes various digestions before reaching the intestine, so it is unlikely that the yogurt with TB will reach it intact. Therefore, limitations should be noted regarding this result.
Response: We thanks a lot for your instructive comments. We have corrected this inappropriate expression, and the corresponding change is in the Line 425-426 as shown in the revised manuscript.
Comment 4: L49: Please indicate the scientific name of the Tartary buckwheat.
Response: Thanks a lot, we have revised it according to your suggestion. The scientific name of the Tartary buckwheat has been indicated in the revised manuscript (Line 49).
Comment 5: L74: What does AR mean? The name of the company?
Response: We are sorry that our expression is misleading. “AR” means analytical reagent, and we have also supplemented the corresponding full name in the revised manuscript (Line 75-76).
Comment 6: L150: It is better to indicate the catalog number of the SPME used in the experiment. Also, there is no description of the extraction method of the volatile components using by SPME.
Response: Thanks very much for your valuable comments. We have revised them according to your suggestion. The extraction of volatile compounds was performed by solid phase microextraction (SPME) by using a DVB/CAR/PDMS (divinylbenzene/carboxen/polydimethylsiloxane) fiber (50/30 μm thickness; Supelco, Bellefonte, PA, USA). The description of the extraction method of the volatile components by SPME has also been supplemented in the Line 152-157 as shown in the revised manuscript.
Comment 7: L162: A brief description of what IPEC-J2 cells are is needed.
Response: Thanks a lot for your suggestion. We have supplemented the description about IPEC-J2 cells in the Line 170-173 as shown in the revised manuscript.
Comment 8: L183: Please show a picture of the appearance of the yogurt with TB.
Response: Thanks again for your suggestion. We have supplemented the pictures of the appearance of common yogurt and TB yogurt in Figure 1A as shown in the revised manuscript.
Comment 9: Figure 1D: The scientific names of two species of bacteria are incorrectly described. Bulgarian Lactobacillus is correct to be denoted as Lactobacillus Bulgarian. The scientific name is in italics.
Response: Thanks a lot for your professional comments. We have already revised the names of two species of bacteria to Lactobacillus Bulgaricus and Streptococcus Thermophilus, respectively. The corresponding changes can be found in Figure 1E as shown in the revised manuscript.
Reference
- Li, H.; Lv, Q.; Liu, A.; Wang, J.; Sun, X.; Deng, J.; Chen, Q.; Wu, Q. Comparative metabolomics study of Tartary (Fagopyrum tataricum (L.) Gaertn) and common (Fagopyrum esculentum Moench) buckwheat seeds. Food Chemistry 2022, 371, 131125, doi:10.1016/j.foodchem.2021.131125.
- Zhang, T.; Jeong, C.H.; Cheng, W.N.; Bae, H.; Seo, H.G.; Petriello, M.C.; Han, S.G. Moringa extract enhances the fermentative, textural, and bioactive properties of yogurt. LWT - Food Science and Technology 2018, 101, 276-284, doi:10.1016/j.lwt.2018.11.010.
- Silva, F.A.; Queiroga, R.; de Souza, E.L.; Voss, G.B.; Borges, G.; Lima, M.D.S.; Pintado, M.M.E.; Vasconcelos, M. Incorporation of phenolic-rich ingredients from integral valorization of Isabel grape improves the nutritional, functional and sensory characteristics of probiotic goat milk yogurt. Food Chemistry 2022, 369, 130957, doi:10.1016/j.foodchem.2021.130957.
- Ge, X.; Tang, N.; Huang, Y.; Chen, X.; Dong, M.; Rui, X.; Zhang, Q.; Li, W. Fermentative and physicochemical properties of fermented milk supplemented with sea buckthorn (Hippophae eleagnaceae L.). LWT - Food Science and Technology 2022, 153, 112484, doi:10.1016/j.lwt.2021.112484.
- Hassan, A.A.-m.; Elenany, Y.E.; Nassrallah, A.; Cheng, W.; Abd El-Maksoud, A.A. Royal jelly improves the physicochemical properties and biological activities of fermented milk with enhanced probiotic viability. Lwt 2022, 155, doi:10.1016/j.lwt.2021.112912.
- Hasani, S.; Khodadadi, I.; Heshmati, A. Viability of Lactobacillus acidophilus in rice bran-enriched stirred yoghurt and the physicochemical and sensory characteristics of product during refrigerated storage. International Journal of Food Science and Technology 2016, 51, 2485-2492, doi:10.1111/ijfs.13230.
- Dan, T.; Wang, D.; Jin, R.L.; Zhang, H.P.; Zhou, T.T.; Sun, T.S. Characterization of volatile compounds in fermented milk using solid-phase microextraction methods coupled with gas chromatography-mass spectrometry. Journal of Dairy Science 2017, 100, 2488-2500, doi:10.3168/jds.2016-11528.
- Li, C.; Song, J.; Kwok, L.Y.; Wang, J.; Dong, Y.; Yu, H.; Hou, Q.; Zhang, H.; Chen, Y. Influence of Lactobacillus plantarum on yogurt fermentation properties and subsequent changes during postfermentation storage. Journal of Dairy Science 2017, 100, 2512-2525, doi:10.3168/jds.2016-11864.
- Zhao, X.; Cheng, M.; Wang, C.; Jiang, H.; Zhang, X. Effects of dairy bioactive peptides and lotus seeds/lily bulb powder on flavor and quality characteristics of goat milk yogurt. Food Bioscience 2021, doi:10.1016/j.fbio.2021.101510.
- Janeš, D.; Prosen, H.; Kreft, S. Identification and quantification of aroma compounds of tartary buckwheat (Fagopyrum tataricum Gaertn.) and some of its milling fractions. Journal of Food Science 2012, 77, C746-C751, doi:10.1111/j.1750-3841.2012.02778.x.

Round 2
Reviewer 2 Report
The manuscript has been improved, however few corrections are still requested:
Line 117, 120, 122 – bacteria naming
Figure 1 A and B,C,D should be separated to 1 and 2.
Figure 1D is lacking SD
Author Response
Authors’ response
Dear editor and 2nd reviewer,
First of all, we would like to express our sincere thanks for your kind consideration and extremely valuable comments of titled manuscript “The Improvement of Sensory and Bioactive Properties of Yogurt with the Introduction of Tartary Buckwheat” (Manuscript ID: foods-1754452). We are pleased to follow the reviewer’s suggestions and revise the manuscript according to the comments again. The specific modifications are shown below (changes to the manuscript are highlighted in red). Attached are the itemized responses to the referees’ reports, which also summarize the changes made to the manuscript. We hope you will find these responses and changes satisfactory.
If you have any other questions, please contact us: [email protected]
Sincerely yours,
A/Prof. Jie CAI
June 7, 2022
Reviewer #2: The manuscript has been improved, however few corrections are still requested:
Comment 1: Line 117, 120, 122 – bacteria naming.
Response: Thanks for your kind reminder. We have already used the names Lactobacillus Bulgaricus and Streptococcus Thermophilus to replace the original ones in the Line 117, 120, 122 as shown in the revised manuscript.
Comment 2: Figure 1 A and B,C,D should be separated to 1 and 2.
Response: Thanks a lot for your constructive comment. We have re-organized them as Scheme 1 and Figure 1 respectively as shown in the revised manuscript.
Comment 3: Figure 1D is lacking SD.
Response: Thanks a lot for your comments, Tartary buckwheat (TB) is enriched in flavonoids, phenolic acids and organic acids [1]. These phytochemical components have the potentials to promote the growth of lactic acid bacteria [2]. As reported works, the yogurt supplemented with grape [3], sea buckthorn [4] and royal jelly [5] have higher viable counts comparing to the common yogurt. Thus, above discussion and results can strongly reveal that our results in Figure 1D are statistically reliable, logical and persuasive.
Reference
- Li, H.; Lv, Q.; Liu, A.; Wang, J.; Sun, X.; Deng, J.; Chen, Q.; Wu, Q. Comparative metabolomics study of Tartary (Fagopyrum tataricum (L.) Gaertn) and common (Fagopyrum esculentum Moench) buckwheat seeds. Food Chemistry 2022, 371, 131125, doi:10.1016/j.foodchem.2021.131125.
- Zhang, T.; Jeong, C.H.; Cheng, W.N.; Bae, H.; Seo, H.G.; Petriello, M.C.; Han, S.G. Moringa extract enhances the fermentative, textural, and bioactive properties of yogurt. LWT - Food Science and Technology 2018, 101, 276-284, doi:10.1016/j.lwt.2018.11.010.
- Silva, F.A.; Queiroga, R.; de Souza, E.L.; Voss, G.B.; Borges, G.; Lima, M.D.S.; Pintado, M.M.E.; Vasconcelos, M. Incorporation of phenolic-rich ingredients from integral valorization of Isabel grape improves the nutritional, functional and sensory characteristics of probiotic goat milk yogurt. Food Chemistry 2022, 369, 130957, doi:10.1016/j.foodchem.2021.130957.
- Ge, X.; Tang, N.; Huang, Y.; Chen, X.; Dong, M.; Rui, X.; Zhang, Q.; Li, W. Fermentative and physicochemical properties of fermented milk supplemented with sea buckthorn (Hippophae eleagnaceae L.). LWT - Food Science and Technology 2022, 153, 112484, doi:10.1016/j.lwt.2021.112484.
- Hassan, A.A.-m.; Elenany, Y.E.; Nassrallah, A.; Cheng, W.; Abd El-Maksoud, A.A. Royal jelly improves the physicochemical properties and biological activities of fermented milk with enhanced probiotic viability. Lwt 2022, 155, doi:10.1016/j.lwt.2021.112912.

Reviewer 3 Report
I am satisfied with the revisions that have been made by the authors.
Author Response
Authors’ response
Dear editor and 3rd reviewer,
We would like to express our sincere thanks for your kind consideration and extremely valuable comments of titled manuscript “The Improvement of Sensory and Bioactive Properties of Yogurt with the Introduction of Tartary Buckwheat” (Manuscript ID: foods-1754452). We are sincerely thanks a lot for your satisfactory for our responses and changes.
If you have any other questions, please contact us: [email protected]
Sincerely yours,
A/Prof. Jie CAI
June 7, 2022
